# Home Range and Movement Patterns of Reintroduced White Lions (*Panthera leo melanochaita*) in the Kruger to Canyons Biosphere Reserve, South Africa

**DOI:** 10.3390/ani12152003

**Published:** 2022-08-08

**Authors:** Jason A. Turner, Emma J. Dunston-Clarke, Inger Fabris-Rotelli, Hans de Iongh

**Affiliations:** 1Institute of Cultural Anthropology and Developmental Sociology, Leiden University, 2300 Leiden, The Netherlands; 2Food Futures Institute, School of Veterinary Medicine, College of Science, Engineering and Education, Murdoch University, Murdoch, WA 6150, Australia; 3Department of Statistics, University of Pretoria, Pretoria 0028, South Africa; 4Institute of Environmental Sciences, Leiden University, 2300 Leiden, The Netherlands

**Keywords:** white lion, reintroduction, home range, movement, habitat selection

## Abstract

**Simple Summary:**

White lions are a natural colour variant of the African lion found within certain lion prides in the Greater Kruger Park Region of South Africa. Human factors led to their absence until white lions were reintroduced in 2006. This study provides the first assessment of home range and movement behaviour of white lions as an index of reintroduction success. Home range is the area where an animal spends its time and encompasses all the resources the animal requires to survive and reproduce. The home range size and average distance walked in a day were compared seasonally (wet compared to dry season) and between sexes (male compared to female) for a pride of white lions and a pride consisting of white and tawny (nonwhite) lions. Both prides had similar sized home ranges, walked a comparable average distance, and preferred similar types of vegetation among which to spend their time (dense woodland compared to open grassland). The white lions from both prides showed natural behaviour, similar to wild lions in terms of how they established and made use of their home ranges, suggesting that white lions can be successfully reintroduced into the wild.

**Abstract:**

White lions are a colour variant of the African lion *Panthera leo melanochaita* and disappeared from the wild due to anthropogenic factors until their reintroduction to the Greater Kruger Park Region of South Africa in 2006. Natural home range behaviour is an index of reintroduction success. Therefore, the home range and movement of a pride of reintroduced white lions and a constructed pride consisting of reintroduced white lions and translocated wild tawny lionesses in small, fenced reserves was assessed. GPS data from collared adults were collected for the white lion pride between 2010–2011 and 2018–2020 for the constructed pride. Home ranges were estimated using kernel density estimation and minimum convex polygon, with minimum daily distance tested for differences between sex, season, and pride. Home ranges were small and average daily movements restricted for both prides (white lion pride: 5.41 km^2^ and 10.44 ± 4.82 km; constructed pride: 5.50 km^2^, 11.37 ± 4.72 km) due to the small reserve size of 7 km^2^. There was no difference between prides for annual and seasonal home range size, male and female home ranges, minimum daily distance travelled, or habitat selection. White lions from both prides established territories and displayed natural home ranging behaviour, suggesting that their reintroduction was successful, in the absence of anthropogenic threats.

## 1. Introduction

Globally, wildlife populations are rapidly declining mainly due to human population growth and consumption [1]. Large carnivores are significantly impacted by human activity due to their ecological requirements and high conflict with humans [2,3]. The increase in human populations worldwide has led to a higher incidence of conflict with many carnivorous species, resulting in population decline and local extinctions [4,5]. The lion has experienced the greatest reduction in range since historic times out of all vertebrate species [6]. Lions have recently been split into two subspecies, *Panthera leo leo* [7], made up of the cats living in Asia and West, Central, and North Africa and *Panthera leo melanochaita* [8], consisting of the lion populations living in East and Southern Africa [9]. This reclassification is necessary to regionally ascertain the threat level of the respective subspecies of lions. In South Africa, lions have disappeared from most of their historical range [10,11] due to habitat loss, human–lion conflict, poaching, and trophy hunting [12]. There are only three surviving historical populations at the time of this study, namely Kruger National Park (KNP) (approximately 1700 individuals) [13], Kgalagadi Transfrontier Park (KTP) (approximately 125 individuals in the South African section) [14,15], and the Greater Mapungubwe Transfrontier Conservation Area (GMTFCA) (<50 individuals) [16]. Since the 1990s, a number of wild but managed lions have been translocated to smaller fenced reserves (<1000 km^2^) across South Africa, totalling approximately 800 lions across 49 fenced reserves [17] and they are managed as a metapopulation [11,18]. The Red Data List for Mammals in South Africa, Lesotho, and Swaziland lists the conservation status of lions in South Africa as Least Concern but indicates that this species would technically qualify for Near Threatened status if the managed subpopulations in small reserves were excluded [12]. These lion subpopulations are therefore a significant contribution to the conservation of lions in South Africa, forming part of the metapopulation management approach, whereby lions need to be regularly translocated to prevent inbreeding, genetic drift, and overpopulation [11].

The white lion is a rare genetic variant of the southern subspecies of the African lion (*Panthera leo melanochaita*) that has a white coat colour and either brown, blue, or green eyes, and has occurred in the Greater Timbavati Region and Central Kruger Park Region of South Africa, since 1938 [19]. The white coat colour is not due to albinism, but rather leucism resulting from a double recessive allele or gene [20,21,22]. White lions and many tawny lions carrying the recessive gene were removed from the wild and placed into captive breeding and hunting progammes, zoos, and circuses worldwide [19,23,24,25]. The anthropogenic impact of lion culling in Central Kruger National Park [26], trophy hunting, and removal from the wild in the Greater Timbavati Region [19,24,25] led to an absence of white lions in their natural habitat up until 2006. The recessive gene was still present in the wild population and white lion cubs were born into the Greater Timbavati Region in 2006–2009, 2011, 2014, 2015, 2018, and 2019, and in Central Kruger National Park in 2014 and 2015 [25]. Only three of the 17 cubs had survived at the time of the present study due to illegal removal to breeding/hunting centres, the continued lion trophy hunting of pride males which caused infanticide, and high impact ecotourism leading to undue stress on lionesses with young cubs during regular viewing by tourist or lodge vehicles [27,28,29]. The natural high mortality rate of 50% of lion cubs within the first year [30] would also have contributed to the low survival rate of white lion cubs between 2006 and 2019. The birth of white cubs in the Greater Timbavati Region and the Central Kruger Park between 2006 and 2019 is confirmation that the recessive gene was still present in the wild population [29]. However, the distribution of the white lion gene in the Greater Timbavati Region and the Central Kruger Park Region has never been determined, nor the frequency of occurrence of the white lion gene within their natural habitat. This is beyond the scope of the present study. At the initiation of this study, there were only three free-roaming white lions within their natural habitat, largely nomadic, and none of them adults. Therefore, our study focused on a reintroduced pride of captive-origin white lions, and a free-roaming pride of captive-origin white lions constructed with wild tawny lions at the Tula Tsau Conservation Area. Turner et al. (2015) [25] showed that there was no difference in the hunting success of the two reintroduced white lion groups compared to wild tawny lions in the same study area, Madjuma Lion Reserve (MLR), Karongwe Game Reserve (KGR), Welgevonden Game Reserve (WGR), Makalali Game Reserve (MGR), and the Associated Private Nature Reserves (APNR) in South Africa. In a recent study, Turner et al. (2022) [31] compared the social behaviour of the reintroduced pride of captive-origin white and wild tawny lions to two captive-origin prides in Zambia and Zimbabwe respectively, and wild tawny lion prides at the Greater Makalali Private Game Reserve in South Africa. The study concluded that the white and tawny lions formed a socially cohesive pride that was more strongly bonded than either the captive-origin or wild tawny prides, a sign of successful reintroduction. In addition to the hunting success and social cohesion of a lion pride, natural home range behaviour is also an index of reintroduction success, and in the case of the white lions, an indication of whether white lions would survive in the wild in the absence of anthropogenic threats. The present study is therefore the first study to investigate the home range, movement patterns, and habitat preferences of white lions, as an index of whether reintroduced white lions show natural home range behaviour.

### 1.1. Home Range and Movements of Lions

A home range is generally defined as the area where an animal acquires resources, mates, reproduces, and takes care of its offspring [32]. Lions are the only truly social cats, generally living in prides [33]. The home range of individual lions is typically confined by the pride’s territory [34]. The ranging behavior of lions is not only dependent on resource availability but also on prey availability, social interactions, habitat quality and reproductive status [35,36,37,38,39]. Lion home range is directly related to prey abundance and the presence of water, thus lower availability of both resources corresponds with larger home ranges and vice versa [36,40,41]. However, other factors, such as social status, sex, age, season, and disturbance (e.g., human population growth and presence of livestock near reserves), may influence the home range [34,42,43]. Group size and territoriality are social factors that also influence home range size [44], with range size increasing with group size [36]. Home ranges are generally smaller in high quality habitats that have a high prey density, such as the Serengeti Plains, compared to low quality or sparse habitats with low prey availability, such as the semi-arid savanna of the Hwange National Park or the dry savanna region of the Central Kalahari in Botswana [42,45,46]. Since rainfall determines habitat quality, structure, and density, through its influence on vegetation health, mediated through edaphic and topographic/catenary gradients, it is a key factor affecting home range size [47,48]. In arid regions, lions adopt larger home ranges to locate prey, which normally occur at low densities. Stander (1991) [49] recorded lion prides with home ranges of up to 2075 km^2^ in Etosha National Park, Namibia, and Funston (2011) [15] documented lion home ranges over 4500 km^2^ in the Kgalagadi Transfrontier Park, South Africa/Botswana. In more mesic habitats, higher prey densities result in smaller home ranges. Home ranges as small as 45 km^2^ were recorded for lions in the Ngorongoro Crater, Tanzania [50]. In wildlife areas that have a distinct wet and dry season, seasonal variation in home range size may therefore occur, such as in the Central Kalahari Game Reserve in Botswana [46].

Lioness home range is determined by suitability for having and caring for young and the availability of sufficient prey to sustain their offspring [42,51]. Male home ranges are influenced by the distribution of females. Therefore, they are generally larger than female home ranges [30,36,42] and can span across numerous prides (Hunter 1998) [52]. Adult male lions maintain a territory largely contiguous with that of their home range [36,53,54]. Large home ranges overlap extensively with those of adjacent prides, while small ranges tend to have little overlap [36].

Lions are known to move up to 20 km in 24 h and to cover hundreds of kilometres over several months [55]. The core of an animal’s home range is defined as the most intensively used area within that animal’s home range [56]. It can be quantified by a utilization distribution that describes the frequency distribution of the locations of an animal at the landscape level [57]. The 50% utilization distribution is often viewed as the core and most important area within the entire home range [56].

### 1.2. Home Range and Movement Patterns of White Lions Compared to Tawny Lions

Although white lions have occurred naturally in the Greater Timbavati Region since 1938 [19] and in the Central Kruger Park Region since 1956 [20,21,26], the home range and movement patterns of white lions have never been studied to determine if they differ from those of tawny lions. Since white lions are the result of a recessive gene [20,21,22], there is a perception that white lions would not survive in the wild, and that their behaviour may not be consistent with that of wild tawny lions. Turner et al. (2015) [25] showed that there was no difference in the hunting success of white lions compared to tawny lions under managed free-roaming conditions, and the social behaviour of a constructed pride of white and tawny lions was similar to that of a pride of wild tawny lions (Turner et al. 2022) [31]. However, home range behaviour and movements of white lions has not been previously published. Home range studies have been conducted on tawny lion prides within smaller reserves in Southern Africa: a private reserve in Gweru, Zimbabawe (1.6 km^2^) [58], Dambwa Forest in Livingstone, Zambia (2.9 km^2^) [57], Madjuma Lion Reserve (15 km^2^) [6], Karongwe Game Reserve, (80 km^2^) [59], Dinokeng Game Reserve (185 km^2^) [60], and Greater Makalali Private Game Reserve (250 km^2^) [58,61] in Limpopo Province, South Africa. Dunston et al. (2017) [58] conducted the first assessment of the spatial ecology and territorial behaviour of captive-origin lions. The study concluded that the two captive-origin prides established territories and core areas in a similar way to a wild pride in the Greater Makalali Private Nature Reserve, South Africa.

This study presents the first investigation into the home range and movement patterns of white lions. Since Hayward et al. (2008) [62] concluded that the use of fences did not affect the natural ranging behaviour of predators, with prey abundance still being the key factor determining space use in fenced reserves, the home range and movement patterns of a reintroduced white lion pride were therefore compared to those a free-roaming pride of white lions integrated with wild tawny lions in the same fenced study area (during different study periods), as well as the two prides of captive-origin tawny lions, and two wild prides studied by Dunston et al. (2017) in fenced reserves [58]. Our study area was 7 km^2^, a similarly small size to the Madjuma Lion Reserve (15 km^2^) in Limpopo Province (South Africa), a fenced reserve that has resident wild tawny lions (see Section 2.1 below for more detail on the Study Area). Our study addressed the following research questions: (i) What is the average home range size of the white lion pride and constructed pride of white and tawny lions? (ii) What is the average distance traveled within the 12 h per day monitoring period by the white lion pride and the constructed pride of white and tawny lions? (iii) Which habitat type is preferred by the white lion pride and constructed tawny lion pride? and (iv) Did the white lion pride and constructed pride of white and tawny lions show natural home range behaviour, movement patterns, and habitat selection similar to that of wild tawny lions, suggesting that the reintroduction to their natural habitat was successful?

## 2. Materials and Methods

### 2.1. Study Area

The study was conducted at the Tula-Tsau Conservation Area (7 km^2^), a fenced wildlife area in the Kruger to Canyons Biosphere Reserve, Limpopo Province, South Africa (Figure 1). The Tula-Tsau Conservation Area is located in the Lowveld of South Africa, and forms part of an important buffer area, the Greater Kruger Environmental Protection Zone (GKEPZ), between the Kruger National Park, rural communities, and the semi-urban town of Hoedspruit. The Tula-Tsau Conservation Area is situated between latitude 24.374° S–24.396° E and longitude 31.106° S–31.146° E. The region experiences a dry winter season (April–September) and a wet summer season (October–March), with a mean annual rainfall of 634 mm.

The Tula-Tsau Conservation Area is a natural Lowveld vegetation area that is classified as Arid Lowveld [63]. The topography is undulating with plains, woodlands of various densities, thickets, and riverine vegetation (Figure 2). This wildlife area is characterised by indigenous flora and fauna, including a wide variety of mammalian prey species: Blue wildebeest *Connochaetes taurinus*, Eland *Taurotragus oryx*, Burchell’s zebra *Equus quagga*, Greater kudu *Tragelaphus strepsiceros*, Nyala *Tragelaphus angasii*, Waterbuck *Kobus ellipsiprymnus*, Common warthog *Phacochoerus africanus*, Bushbuck *Tragelaphus sylvaticus,* Impala *Aepyceros melampus*, Steenbok *Raphicerus campestris*, and Common Duiker *Sylvicapra grimmia*. The following small carnivores are common in the area: Black-backed jackals, *Canis mesomelas*, and caracal *Caracal caracal*, whilst these larger carnivores are often seen but are not resident: leopard, *Panthera pardus*, spotted hyaena *Crocuta crocuta*, African wild dog *Lycaon pictus*, and serval *Leptailurus serval*. A single lion pride occurred at the Tula-Tsau Consevation Area: a pride of white lions from 2007 to 2013 (which were translocated to another wildlife area), and a pride of white and tawny lions from 2014 to 2020. Wild lions are present on the neighbouring wildlife areas of Kapama, Thornybush, and Timbavati Private Nature Reserve. The prey population has been replenished on an annual basis since the Tula Tsau Conservation Area is a small fenced reserve. Prey abundance and composition were therefore consistent for the period of study.

The white lion pride and the constructed pride of white and tawny lions both had a similar pride structure with one or more male lions, two or more adult lionesses, and several subadults or cubs (Table 1).

### 2.2. Data Collection and Sampling

The Royal pride was a reintroduced pride of white lions that comprised of an adult male, one adult female, two subadult males, and a subadult female (Table 1). The constructed pride (Tsau) of white and tawny lions comprised of two adult white lion males, two adult tawny females, two subadult tawny males, and a subadult tawny female (Table 1). All adult lions in each pride were fitted with Africa Wildlife Tracking (AWT, Pretoria, South Africa) VHF radio collars: two adults for the Royal pride (40%), and four adults for the Tsau pride (60%). Data were collected for the Royal pride for 2010 and 2011, and for the constructed white and Tsau pride between 2018 and 2020. Average rainfall was similar for the data collection periods: 655 mm for the 2010–2011 period and 608 mm for the 2018–2020 period. The average rainfall during the two study periods was also similar to the long term mean annual rainfall of 634 mm (2000 to 2020). Since there were no significant differences in climate, habitat type, prey composition or prey abundance between the two study periods (2010/2011 and 2018/2020), therefore time difference between pride monitorings was unlikely to impact results. Using VHF radio telemetry (Communication Specialist R-1000 Receiver) and a Garmin eTrex GPS, the location and movement of the reintroduced prides were recorded daily during their peak activity periods; from dusk to dawn (1700 h to 0500 h). This was done during the following periods; for the white lion pride: 25 February 2010 to 12 October 2011; and for the constructed pride of white and tawny lions: 1 January 2018 to 31 December 2020. Additional location and movement data were recorded during the study periods by vehicle for direct observations of their hunting ability and hunting behavior, as described in Turner et al. (2015) [25].

### 2.3. Data Analysis and Statistical Testing

The following statistical software packages of the program R [64] were used for data analysis and statistical testing: sf, spatstat, maptools, sp, leaflet, mapview, openair, ks, ggplot2, ggspatial, raster, rgdal, and cleangeo [65,66,67,68,69,70,71,72,73,74,75,76,77].

For the first research question, we calculated the annual and seasonal home range of each collared lion by calculating the minimum convex polygon (MCP) and kernel density estimation (KDE) home ranges, using the program R [64]. MCP is a commonly used, simple measure chosen to allow comparison with previous home range estimates. This method calculates the smallest convex polygon from all the locations available: 100% MCP for all lion locations, and 95% MCP which removes 5% of the outliers in the dataset. A more recent method is KDE, which is based on density estimations of GPS locations, by calculating the harmonic means and creating isopleths of intensity of home-range utilization [56]. The boundary of the lion’s home range is 95% KDE, and the core home range is 50% KDE [78]. Since animals utilise areas within their home range unevenly, fixed kernel density estimators were used to define the core areas [79]. For each home range (i.e., 50% KDE, 95% KDE, and 100% MCP) a mixed effect model was built, with sex (male/female), season (wet/dry), and pride (Royal/Tsau pride) as explanatory categorical variables. Contrary to home range studies of wild tawny lion prides, such as the work of Lesilau et al. (2021) [80], male lions were not excluded from pride home range estimators since males were not involved in pride takeovers due to the absence of other lion prides in the small fenced (closed) system. Statistical analyses were performed using the program R [64].

To answer the second research question, the minimum potential daily distance traveled was calculated as the Euclidean distance between following GPS fixes within 12 h based on the method described by Hunter (1998) [52]. In contrast to Hunter (1998) [52], a 12 h period rather than a 24 h period was used, due to the logistical and resource limitations of our study, and since the 12 h period was during the peak activity period for lions in the Tula Tsau Conservation Area, between dawn and dusk (1700 h to 0500 h) [25]. The selection of the 12 h observation period (1700 h to 0500 h) for our study is supported by the observations of Hayward and Slotow (2009) [81] that lion activity peaked between 2100 h to 2400 h and 0200 h to 0700 h, as well as Schaller (1972) [34] who observed that lions are active post 1700 h, and Rudnai (1973) [82] that lion activity increases post 1600 h. A mixed effect model was built with daily distance traveled as response variable, and with sex and season as defining categorical variables.

For the third research question, a vegetation class was assigned to each GPS fix to determine the proportion of fixes in each habitat type. The vegetation types for Tula Tsau Conservation Area were described by Mcdonald (2005) [83]. Manly’s selection index was used to assess lion habitat preference [84], as described by Lesilau et al. (2021) [80]. The selection index was measured using the formula: wi = oi/pi, where wi = selection index for vegetation type i; oi = proportion (number) of fixes in vegetation type i; and pi = proportion of vegetation i available in the park. Values above 1.0 indicated preference, while values less than 1.0 indicated avoidance. The standardized index Bi allowed comparisons between habitat types: Bi = wi/(Σni = ŵi). Values below 0.250 (corresponding to 1/number of vegetation types) indicated relative avoidance, while values above indicated relative preference.

For the mixed effects models, a likelihood ratio test (LRT) was calculated with a Chi-squared test. The mixed effect models were built using the function lmer() from the package lmerTest [85]. Lion identity and year were included as random factors for each model. Response variables for the mixed effect models met model assumptions of normality and homoscedasticity.

## 3. Results

### 3.1. Home Ranges and Movements

The individual home range size for the members of the Royal pride and Tsau pride are shown in Table 2 and Table 3. The home range size was similar for the individuals of each pride, and between prides, for all sexes, ages, and seasons (Table 2, Table 3, Table 4, Table 5 and Table 6).

The Royal pride and Tsau pride had home ranges that extended over the majority of the reserve (77% and 80%), with the Royal pride occupying 5.41 km^2^ (95% KDE) and the Tsau Pride 5.50 km^2^ (95% KDE) within the 7 km^2^ reserve (Figure 3). Kernel densities indicated areas most used by each pride. The Royal pride and Tsau pride both used seven core areas (50% KDE), four of them situated around a waterhole for the Royal pride, and five for the Tsau pride (Figure 3). The mean annual home range size of lions in the Tula Tsau Conservation Area was 5.53 ± 0.35 km^2^ (95% KDE). Annual core home ranges (50% KDE) were on average 1.76 ± 0.10 km^2^ in size, which was around 32% of the 95% KDE home range. Using 100% of the GPS fixes resulted in a mean annual home range size of 5.70 ± 0.23 km^2^ (100% MCP), covering 81% of the Tula Tsau Conservation Area.

The mean annual home range size of the prides at the Tula Tsau Conservation Area (5.53 ± 0.35 km^2^) did not differ significantly from the mean seasonal home range size (5.33 ± 0.34 km^2^) (95% KDE; X^2^ = 2.561, df = 1, *p*-value = 0.109). No difference in size was found between wet season home ranges and dry season home ranges for the Tula Tsau Conservation Area (Table 5; Figure 4). Seasonal home ranges (95% KDE and 100% MCP) for males and females did not differ significantly in size, or between prides (Table 5) (Figure 4, Figure 5 and Figure 6). The home range size for the white lion males of the Royal pride was similar to that of the white lion males of the Tsau Pride (Table 2 and Table 3; Figure 5). Core home ranges (50% KDE) were consistent for sex, season, and pride (Table 4, Table 5 and Table 6).

The average potential minimum distance traveled within 12 h by lions from the Tula Tsau Conservation Area was 10.91 ± 4.90 km. There was no significant difference between the average distance traveled by the Royal pride (10.44 ± 4.82 km) and the Tsau pride (11.37 ± 4.72 km) (X^2^ = 0.719, df = 1, *p*-value = 0.290). Males and females from both prides traveled similar distances (Table 4; Figure 4), with a maximum of 19.30 km traveled in 12 h by male M and female T from the Tsau pride. The distance traveled by lions at the Tula Tsau Conservation Area was not affected by season (Table 5; Figure 4).

A summary of home range size and movement metrics for the different variables tested is given in Table 4, Table 5 and Table 6 and Figure 4.

### 3.2. Habitat Selection

The habitat analysis showed that the most preferred habitat selected by both the Royal pride and the Tsau pride was thickets (Table 7). This habitat type was the largest habitat type at Tula Tsau Conservation Area. The Royal pride also showed a preference for woodland habitat, avoiding plains. The Tsau pride showed some preference for woodland, but similarly avoided plains habitat. Riverine habitat was the second largest habitat type but was avoided by both the Royal and Tsau prides.

## 4. Discussion

Establishing a home range is important for a territorial species and is therefore indicative of short-term reintroduction success. The Royal and Tsau prides occupied the Tula-Tsau Conservation Area at different times, but both established home ranges and had defined movement patterns within the fenced reserve. The home range behaviour between both prides was similar. Ranging dynamics for the study prides were compared to that of previously studied captive-origin and wild tawny prides.

The home range dynamics of the reintroduced white lion pride (Royal pride) was similar to that of the constructed pride of white and tawny lions (Tsau pride) at an individual and a pride level, with a home range that extended over the majority of the reserve (Figure 3). The difference in the study period for the two prides is unlikely to have had an impact on the results since both prides existed on the same reserve, which had similar climatic conditions, prey composition, and prey abundance across all studied years. Average rainfall for both study periods was comparable and similar to the long-term average rainfall calculated between 2000 and 2020. Being a small fenced reserve with predators, including lions, spotted hyaena, and leopard, the prey population is replenished on an annual basis and maintained at the ecological carrying capacity of the reserve. Dunston et al. (2017) [58] similarly found captive-origin tawny lion prides to establish home ranges within small reserves in Zambia and Zimbabwe. The first of these prides occupied a 1.5 km^2^ territory within a 1.7 km^2^ reserve (76%) and the other pride had a 2.2 km^2^ territory within a 2.9 km^2^ reserve (88%), respectively. The reintroduced white lion pride and constructed pride of white and tawny lions therefore established a home range in a similar way to these previously studied captive-origin tawny prides.

In larger or unfenced reserves in South Africa with a similar habitat type and prey density, wild lion prides establish larger home ranges that do not typically extend over the majority of the reserve, as was observed for wild tawny lion prides at the Phinda Resource Reserve (KwaZulu Natal), and Greater Makalali Private Nature Reserve (Limpopo Province) [52,58]. In accordance with wild prides in other studies, the Makalali pride established larger home ranges (28.5 km^2^ and 56 km^2^ within a 234.8 km^2^ reserve) extending over only 12 to 24% of the reserve [58], based on prey density and factors that reflect prey availability, such as environment and season [38,39,42,55,86]. Similar to the captive origin-prides studied by Dunston et al. (2017) [58], the Royal and Tsau prides showed signs of natural ranging behaviour within the limited available area. Although the reserve size, and consequential lion home range size, for the Royal and Tsau prides were significantly smaller than the estimated territory sizes of wild prides; range of 50 to 7400 km^2^ [41,42,55,86], and mean territory size of 56 km^2^ (range of 15–219 km^2^), nearly 20% of the wild lion population in South Africa is protected within 49 smaller fenced reserves, with several of them being significantly smaller than the Greater Makalali Private Game Reserve [11]: Mabula Game Reserve (16.5 km^2^), Thanda Private Game Reserve (70 km^2^), Karongwe Game Reserve (79 km^2^), Thornybush Nature Reserve (116 km^2^), and Shamwari Game Reserve (139 km^2^) [11]. In many of these South African reserves, interventionist conservation management of territorial large carnivores has taken place, where farmland has been rehabilitated to game reserves and many species were reintroduced [87,88,89]. A restricted reserve size means the lion pride(s) need to be intensively managed to ensure overpopulation and inbreeding of lions does not occur, that a balance between all predator populations exists and that prey populations are not depleted. Regular translocation and manipulation of pride structure are often necessary (e.g., replacement of pride males with new males to promote genetic diversity or breeding control of females), disrupting the pride social structure [90,91]. In fenced reserves that have more than one pride present, the smaller the reserve, the higher the intraspecific and interspecific competition, territorial stress, and the more likely that lions may break out the reserve into a neighbouring property or community-owned land, potentially becoming damage-causing animals (i.e., a threat to livestock and human life) [43,92]. A better understanding of home range behaviour and dynamics of lions in smaller reserves is therefore important for successful pride reintroduction and management. This is particularly important for the management of white lions, which were extirpated due to anthropogenic factors and are still under threat due to trophy hunting in parts of their natural habitat, meaning intensive management of these prides will be required for the near future.

The home range size of the pride males and lionesses for the Royal and Tsau prides were not significantly different due to the small and limited reserve size, and the fact that resource availability and access to females for both prides were satisfied by occupying the majority of the reserve. Abundant food and high-quality habitat allow an animal to meet its biological requirements in a relatively small home range [35,93]. Home ranges of male lions are often larger than those of females and may encompass two or more female prides [30,52,94,95]. The home range size and establishment of a territory by wild prides may be influenced by the presence of other prides and male coalitions [34], as has been observed for wild prides in Kruger National Park [36], Phinda Resource Reserve [52], Welgevonden Private Game Reserve [96], and the Greater Makalali Private Nature Reserve [58]. The Royal and Tsau prides were not influenced by other prides or male coalitions since the Tula Tsau Conservation Area was not large enough to support more than one pride. Although pride males and lionesses for both prides utilized the majority of the available home range, the males were often located at the reserve fence boundary in response to the territorial roaring and presence of pride males on the neighbouring Kapama, Thornybush and Timbavati Private Nature Reserves. Dunston et al. (2017) [58] had similar observations for the males from captive-origin prides, which were regularly located close to the reserve boundary, in response to the external stimuli of large game species and roaring of lions on the other side of the boundary fence. We therefore postulate that the white lion males from the Royal and Tsau prides may establish larger home ranges in a similar way to wild lions, if the reserve size was larger. In support of our postulation, Hayward et al. (2008) [62] concluded that the use of fences has not affected the natural ranging behaviour of predators, with prey abundance still being the key factor determining space use in fenced reserves. The white lioness from the Royal pride and white lion males from both prides therefore showed natural home range behaviour consistent with adapting to the limited reserve size, suggesting that the reintroduced captive-origin white lions and constructed pride of white and tawny lions made optimum use of the available habitat.

The absence of seasonal variation in home range size for either the Royal or Tsau prides, or between males and females for either pride, is likely a reflection of the abundant prey availability and accessibility year-round due to the small reserve size, annual restocking with prey species, and high availability of water (Turner et al. 2015) [25]. Seasonal variation in the home range size of wild prides has been observed in regions where there is a distinct wet and dry season, such as at the Phinda Resource Reserve in KwaZulu Natal (South Africa), Greater Makalali Private Game Reserve in Limpopo (South Africa), Dinokeng Private Game Reserve in Gauteng and Limpopo Provinces (South Africa), and the Cenral Kalahari Game Reserve in Botswana [46,52,60,61]. However, a lack of seasonal variation in home range size has also been found for wild lion prides [36,39], such as in Nairobi National Park (Kenya) [80], and Majete Wildlife Reserve (Malawi) [97], which are smaller fenced reserves (<1000 km^2^) with a high prey density and availability of water. Although the Greater Makalali Private Game Reserve experienced a similar climate to the Tula Tsau Conservation Area, seasonal variation in home range size was observed for the wild prides in that reserve due to the much larger reserve size and greater incidence of pride dispersal. Our study therefore concluded that both the reintroduced white lion pride and the constructed pride of white and tawny lions showed no seasonal variation in ranging behaviour, which is consistent with natural wild prides in small reserves that have a high prey abundance and availability of artificial water sources.

The distance moved by the Royal and Tsau prides was comparable, and there was no difference in the distance moved by male lions and lionesses for either pride. The limited reserve size could support only one pride and meant pride males did not have to move between prides and females did not have to move significant distances to find potential prey. The significant variability in the size of lion home ranges, ranging between 50 and 7400 km^2^ [41,42,55,86], means that lions are known to move up to 20 km in 24 h and to cover hundreds of kilometres over several months [55]. Although the average distance traveled by the Royal and Tsau prides was calculated over 12 h and not 24 h as in other studies, such as those of Hunter (1998) [80] and Lesilau et al. (2021) [80], the level of activity of these prides during the 12 h observed was comparable to wild prides studied in similar woodland and plains habitats. Hanby et al. (1995) [50] observed inactivity to occur at 79% for a pride in the Ngoronoro Crater (woodland) and 78.5% for a pride on the Serengeti Plains (plains), compared to 81.7% for the Royal pride and 84.1% for the Tsau pride, during the 12 h observation period. The mean and maximum distance traveled by the Royal and Tsau prides (10.91 km; 19.3 km) was comparable to that observed for wild prides in the Nairobi National Park (4.5 km; 29.9 km) [80] and Karongwe Game Reserve (5.4 km; 24.0 km) [59], which also had small home ranges of 14–51 km^2^ (95% KDE) and 35–69 km^2^ (95% KDE), respectively. This is in agreement with the conclusion of Lesilau (2019) [98] that areas of high prey density seem to result in small home ranges and short daily distances traveled by lions and suggests that the average distance traveled for white lions may be similar to that of tawny lions. The reintroduced white lions therefore showed natural ranging behaviour within the available sized reserve.

The habitat preference of the Royal and Tsau prides for thickets and woodland vegetation rather than open plains was not surprising since lions are an ambush predator. Vegetation cover is more significant than type of terrain, since hunting lions usually use vegetation patches to stalk closer to their prey, or they wait hidden until their prey is sufficiently close to attack [99,100]. The slight difference in preference for the selected habitat types by the Royal and Tsau prides was most likely due to the difference in pride structure, with the Tsau pride having two adult males and females, compared to one adult male and female in the Royal pride. A coalition of pride males typically spends little time with the pride, being away patrolling their territory and hunting more effectively as a coalition [95]. Within Africa, lions are found in a range of habitats from open to closed woodland and have adapted to survive in the arid outskirts of deserts to the borders of the dense tropical Congo forest [101]. Many lion studies have found a preference for riverine vegetation [58,62,80], which was not observed in our study. The apparent avoidance of riverine vegetation by the prides in our study was due to research vehicles being unable to access and view the lions when in this vegetation type rather than the prides avoiding use of this habitat. The proximity to natural or artificial water sources has been found to be an important factor in lion ranging behaviour. The provision of artificially supplied water during the dry period in wildlife reserves may affect the movement patterns of ungulates [102,103,104], and as a result the distribution and ranging patterns of predators such as lions. This was evident at the Tula Tsau Conservation Area, where the home range of the Royal and Tsau prides were centred around the high number of artificial and natural water sources. Dunston et al. (2017) [58] had similar observations for the captive-origin prides in Zimbabwe and Zambia, and the wild prides at the Greater Makalali Private Game Reserve. The availability of water and consequential high prey abundance have been found to be significant factors in the home range behaviour of wild lion prides at Phinda Resource Reserve, Karongwe Game Reserve, Central Kalahari Game Reserve, and Nairobi National Park [46,52,59,80]. The findings for the habitat preference and home range behaviour of both the prides in our study therefore show similarities to those for wild prides, and accordingly suggest that the home range behaviour of white lions seems to be similar to that of tawny lions.

The natural home range behaviour of the Royal and Tsau prides post-release was an indication of their reintroduction being successful. We believe that the movements and home ranges of these lions may also have been influenced by the fenced areas in which they are kept. We postulate that in the absence of anthropogenic threats (artificial removal to breeding camps, culling and trophy hunting), white lions are capable of surviving in their natural habitat. This is supported by the fact that captive-origin white lions have been determined to show similar hunting success and social behaviour to wild tawny lions [25,31], and wild white lions have been observed to survive successfully and reproduce in their natural habitat [19,23,24,29].

A number of limitations exist in the present study: namely the small size of the fenced reserve, prides, and sample size, and the difference in the time periods covered. At the time of our study, there were no adult white lions in the wild, a future study is therefore recommended when the prides in the Timbavati Private Nature Reserve and Kruger National Park have adult white lions. The home range behaviour of the white lion pride should ideally have been compared to that of a wild tawny pride, and not a constructed pride of white and tawny lions. Future studies should compare the home range behaviour and movements of larger prides of free-roaming white and tawny lions in big reserves or ecosystems that are at least the mean home range size for wild lions (56 km^2^), and ideally within the open system of the Timbavati Private Nature Reserve or Kruger National Park. The living circumstances of the prides and vegetation differences also need to be considered when conducting comparisons between captive-origin and wild prides. We did not have a sample of truly independent samples and the conclusions should therefore be viewed as preliminary. Hence, further work in this area is recommended.

## 5. Conclusions

The present study suggests that the home ranging behaviour and movement of white lions was similar to that of tawny lions within small fenced reserves. This indicates that the reintroduction of captive-origin white lions into their natural habitat was successful, despite the limited reserve size. The reintroduction of white lions at the Tula Tsau Conservation Area is important in securing a protected subpopulation since the anthropogenic threat of trophy hunting still exists in parts of the species’ natural range. This study provides critical information for the metapopulation management of lions and informs the use of reintroduced and constructed prides for lion conservation.

## Figures and Tables

**Figure 1 animals-12-02003-f001:**
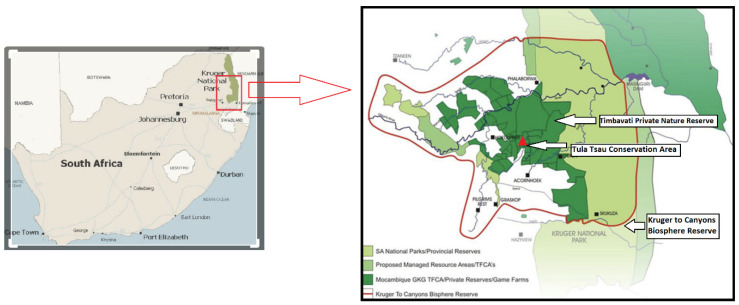
Location of the Tula-Tsau Conservation Area, Kruger to Canyons Biosphere Reserve, South Africa.

**Figure 2 animals-12-02003-f002:**
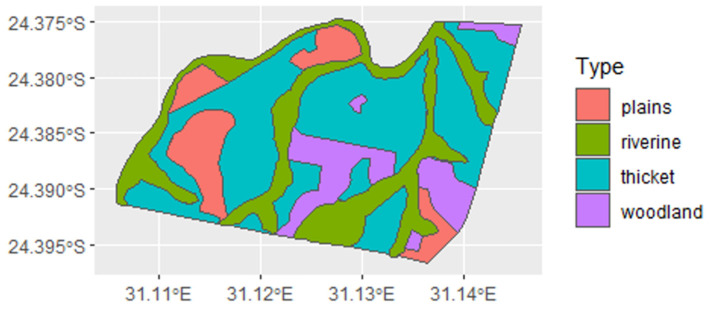
Vegetation map of the Tula Tsau Conservation Area Limpopo, South Africa.

**Figure 3 animals-12-02003-f003:**
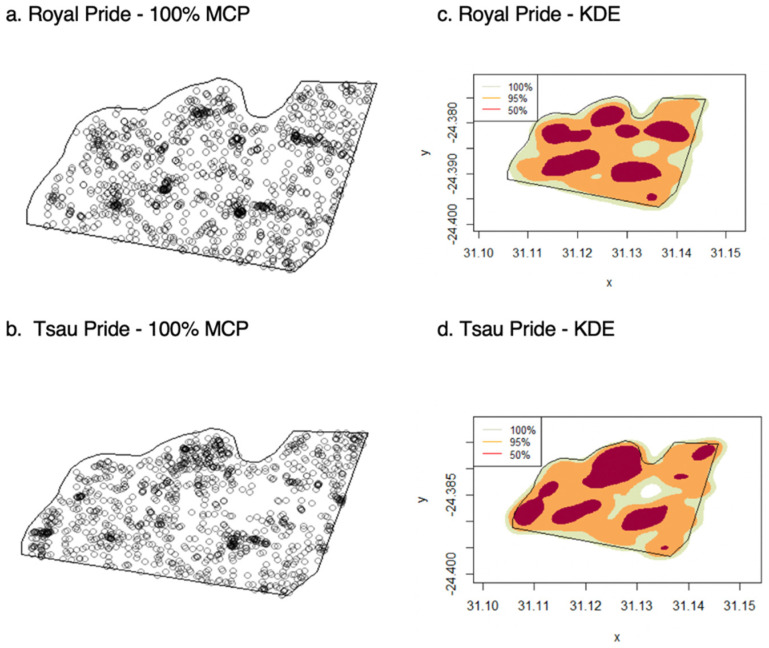
Recorded locations and home range distribution for the Royal and Tsau prides, using 100% MCP (**a**,**b**), and KDE(100%, 95% and 50%) (**c**,**d**).

**Figure 4 animals-12-02003-f004:**
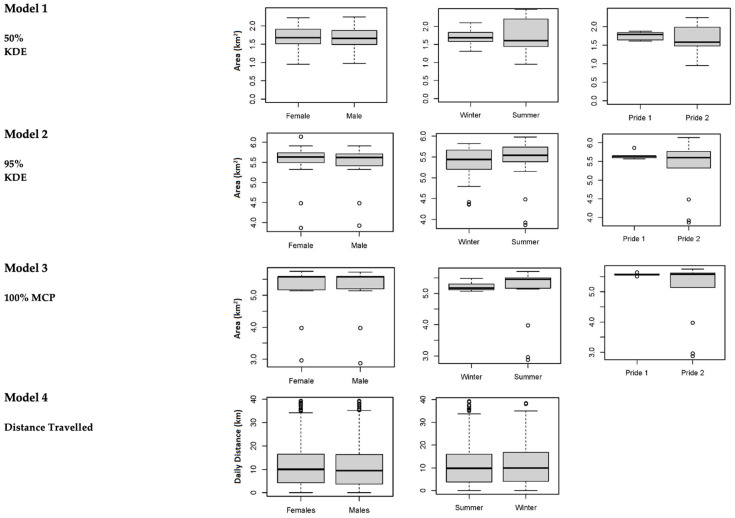
Differences in home range size and movement metrics for the different variables tested for lions at the Tula Tsau Conservation Area, between 2010 and 2011 (Royal Pride) [Pride 1] and 2018 and 2020 (Tsau Pride) [Pride 2]. Box plot method was used; showing upper and lower limits, the median value, and open circles indicate outlier values.

**Figure 5 animals-12-02003-f005:**
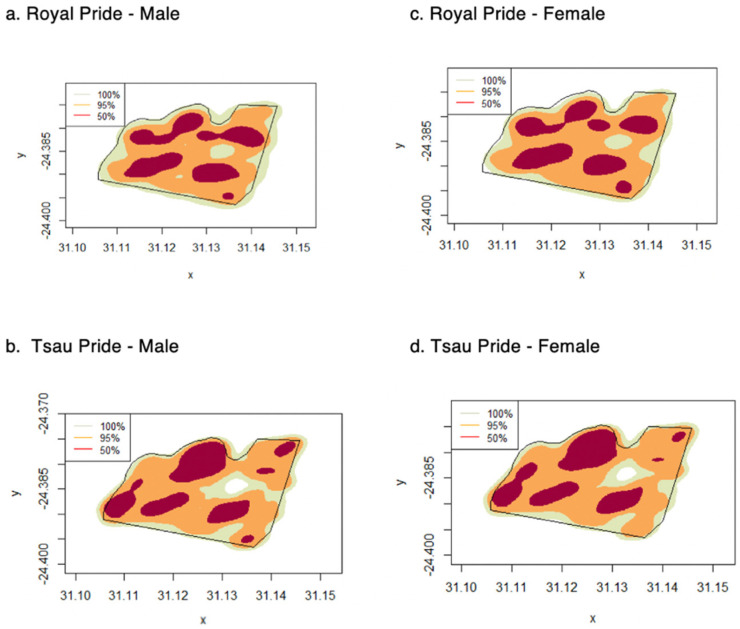
Home range distribution for male and female lions for the Royal and Tsau prides, using KDE (100%, 95% and 50%) (**a**–**d**), Tula Tsau Conservation Area, Limpopo, South Africa.

**Figure 6 animals-12-02003-f006:**
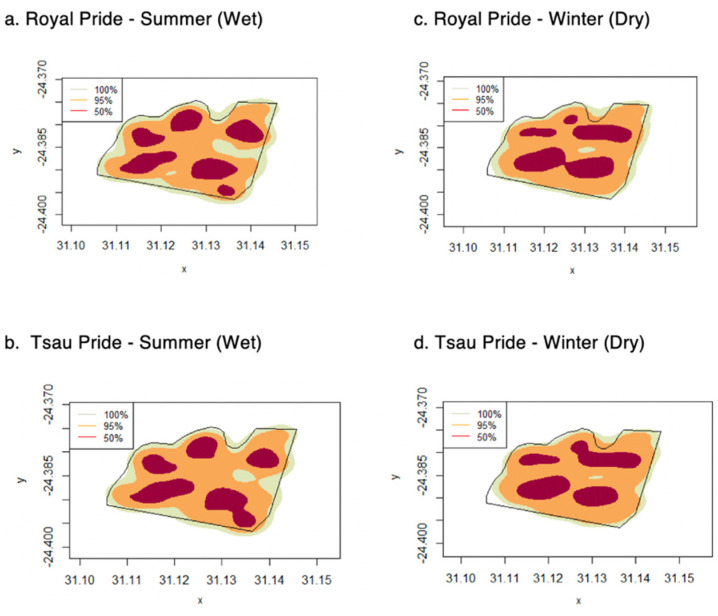
Seasonal home range distribution for the summer (wet) and winter (dry) seasons, for the Royal and Tsau prides, using KDE (100%, 95% and 50%) (**a**–**d**), at the Tula Tsau Conservation Area, Limpopo, South Africa.

**Table 1 animals-12-02003-t001:** Pride composition of the Royal pride and the Tsau pride, at the Tula-Tsau Conservation Area (7 km^2^) (24°23′ S/31°98′ E).

Pride	Adult (>4 yr)	Adult (>4 yr)	Sub Adult (2–4 yr)	Sub Adult (2–4 yr)	Cub (<2 yr)	Cub (<2 yr)	Total	Observation Period
	Male	Female	Male	Female	Male	Female		
Royal	1 *	1	2	1	0	0	5	25 January 2010–12 December 2011
Tsau	2 *	2	2	1	0	0	7	1 January 2018–31 December 2020

* Indicates an individual who is of captive-bred origin.

**Table 2 animals-12-02003-t002:** Summary of individual home range size per lion (100% MCP, 95% KDE; 50% KDE), for the Royal pride, at the Tula-Tsau Conservation Area, South Africa.

Lion	MCP	KDE 95%	KDE 50%
Male 1 (RM1)	6.81	5.47	1.69
Female 1 (RF1)	6.33	5.32	1.59
Subadult Male 1 (RSM1)	6.31	5.21	1.58
Subadult Male 2 (RSM2)	6.31	5.21	1.61
Subadult Female 1 (RSF1)	6.31	5.21	1.61

**Table 3 animals-12-02003-t003:** Summary of individual home range size per lion (100% MCP, 95% KDE; 50% KDE), for the Tsau pride, at the Tula-Tsau Conservation Area, South Africa.

Lion	MCP	KDE 95%	KDE 50%
Male 1 (TM1)	6.58	5.64	1.62
Male 2 (TM2)	6.89	5.72	1.61
Female 1 (TF1)	6.52	5.53	1.58
Female 2 (TF2)	6.79	5.61	1.61
Subadult Male 1 (TSM1)	6.51	5.37	1.61
Subadult Male 2 (TSM2)	6.51	5.37	1.61
Subadult Female 1 (RSF1)	6.51	5.37	1.58

**Table 4 animals-12-02003-t004:** Annual home range size and movement metrics of lions in the Tula Tsau Conservation Area, South Africa, between 2010 to 2011 (Royal Pride) and 2018 to 2020 (Tsau Pride) (adult males: *n* = 3; adult females: *n* = 3; subadult males: *n* = 4; subadult females: *n* = 2).

Home Range	All Years	Males	Females
50% Kernel Density Estimator (km^2^)	1.76 ± 0.10	1.85 ± 0.31	1.79 ± 0.32
	(1.61–1.87)	(1.48–2.24)	(1.47–2.20)
95% Kernel Density Estimator (km^2^)	5.53 ± 0.35	5.55 ± 0.21	5.50 ± 0.23
	(5.50–5.64)	(5.20–5.75)	(5.14–5.71)
100% Minimum Convex Polygon (km^2^)	5.70 ± 0.23	5.75 ± 0.21	5.63 ± 0.41)
	(5.32–6.25)	(5.41–6.14)	(5.60–5.66)
Daily Distance (km)	10.91 ± 4.90	11.54 ± 5.12	11.31 ± 4.94
	(0.01–19.22)	(0.01–19.30)	(0.01–19.30)

**Table 5 animals-12-02003-t005:** Seasonal home range size and movement metrics of lions in the Tula Tsau Conservation Area, South Africa, between 2010 to 2011 (Royal Pride) and 2018 to 2020 (Tsau Pride) (adult males: *n* = 3; adult females: *n* = 3; subadult males: *n* = 4; subadult females: *n* = 2).

Home Range	All Seasons	Summer (Wet)	Winter (Dry)
50% Kernel Density Estimator (km^2^)	1.83 ± 0.28	1.89 ± 0.39	1.77 ± 0.17
	(1.37–2.47)	(1.37–2.47)	(1.58–2.10)
50% Kernel Density Estimator (km^2^)	5.33 ± 0.34	5.44 ± 0.21	5.21 ± 0.11
	(5.07–5.70)	(5.14–5.70)	(5.07–5.35)
100% Minimum Convex Polygon (km^2^)	5.57 ± 0.20	5.65 ± 0.21	5.48 ± 0.19
	(5.15–5.98)	(5.15–5.98)	(5.20–5.70)
Daily Distance (km)	9.46 ± 4.59	9.62 ± 4.75	9.29 ± 4.41
	(0.01–18.24)	(0.01–18.51)	(0.01–17.96)

**Table 6 animals-12-02003-t006:** Pride home range size of lions in the Tula Tsau Conservation Area, South Africa, between 2010 to 2011 (Royal Pride) and 2018 to 2020 (Tsau Pride) (adult males: *n* = 3; adult females: *n* = 3; subadult males: *n* = 4; subadult females: *n* = 2).

Home Range	Royal Pride	Tsau Pride
50% Kernel Density Estimator (km^2^)	1.76 ± 0.10	1.81 ± 0.29
	(1.61–1.87)	(1.47–2.24)
50% Kernel Density Estimator (km^2^)	5.57 ± 0.35	5.51 ± 0.21
	(5.50–5.64)	(5.14–5.75)
100% Minimum Convex Polygon (km^2^)	5.65 ± 0.82	5.70 ± 0.23
	(5.57–5.86)	(5.32–6.25)

**Table 7 animals-12-02003-t007:** Habitat selection by the Royal and Tsau prides at Tula-Tsau Protected Area (TTPA), South Africa.

Vegetation Type	Proportion of TTPA	Royal Pride	Royal Pride	Tsau Pride	Tsau Pride
		W_i_	B_i_	W_i_	B_i_
Woodland	12%	1.595	0.288	1.052	0.184
Plains	13%	0.239	0.043	0.390	0.068
Riverine	27%	1.193	0.216	1.299	0.227
Thicket	48%	2.502	0.453	2.980	0.521

W_i_: selection index; values above 1.0 indicate preference; values less than 1.0 indicate avoidance. B_i_: standardized selection index, which allows comparisons; values below 0.250 indicate relative avoidance; values above 0.250 indicate relative preference.

## Data Availability

This research and its data forms part of a PhD thesis registered with Leiden University, Leiden, Netherlands.

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
