# Peer review of "Home Range and Movement Patterns of Reintroduced White Lions (Panthera leo melanochaita) in the Kruger to Canyons Biosphere Reserve, South Africa"

_animals, 2022, doi:10.3390/ani12152003_

Round 1
Reviewer 1 Report
Manuscript ID: animals-1828784
Home range and movement patterns of reintroduced white lions (Panthera leo melanochaita) in the Kruger to Canyons Biosphere Reserve, South Africa
Authors: Jason A. Turner *, Emma Dunston-Clarke, Inger Fabris-Rotelli, Hans De Iongh
Review
Very interesting paper, as white lions are rarity, and their reintroduction is only one of the many conservation problems facing Africa. I see potential of the manuscript, however, there are several comments for authors to be implemented into revision.
General question:
Authors wrote: “Stander recorded lion prides with home ranges 130 of up to 2 075 km2 in Etosha National Park, Namibia, and Funston documented lion home ranges over 4 500 km2 in the Kgalagadi Transfrontier Park, South Africa/Botswana. In more mesic habitats, higher prey densities result in smaller home ranges. Home ranges as small as 45 km2 were ...”, and – investigation of the home ranges of two prides in 7 km2 enclosure? I understand, there were limitations, introductions, etc. – but anyway, authors should explain, if their results are at least close to natural pattern.
These issues should be somehow reflected when presenting study aims.
Material and Methods
There are many names of the places presented, and it is not easy to understand how these localities are inter-related. Therefore, authors should present all used place names on the map of Figure 1. E.g., Line 191 says “Tula-Tsau Conservation Area”, but on figure 1 we see “Tsau conservancy Area”. Are these the same? Can these name be unified and used throughout?
Table 1: should we understand, that both prides are living together? GPS is the same.
All “mm”, “km” should be separated from numerical values.
Line 244: packages used in R must be shown and referenced.
Lines 364–265: (17:00 to 05:00) but (17h00 to 05h00) – make this consistent,
Lines 243–244 and 282–283 – repetition.
I still cannot understand, if studied two prides were present in both investigation periods?
Results
Main comment – overlap between pride areas must be shown, and discussed – is this natural, or related to very limited area available for both prides?
Tables 3 and 4 are not needed as there are no differences; it is enough to mention this in the text.
Numeration of figures: Figure 2 should be followed by Figure 3, then Figure 4, etc. Parts of the figures are labelled as (a), (b) – see Template.
Figure 4 – daily distances shown as 5–15 km, but Table 5 indicates distances as being 1.2 km on average, Something should be wrong here.
Chapter Habitat selection: Table 6.1 and 6.2 should be merged into Table 6, as now both tables simply take space.
Control question on Tables 6.1 and 6.2 – if proportions are the same for both prides, why Wi and Bi are different? Would be nice to explain in the text.
Discussion
Line 344: “The Royal and Tsau prides both established home ranges and had defined movement patterns within their fenced reserves.” – but it was the same territory, as I understand.
If pride occupies 80% of the fenced area, this is far from being natural, and I would like authors to express this in the text.
Citations
Should be presented according the Template, e.g., [1,3] not [1, 3], and [91–93] not [91-93].
Back matter
Line 527: delete “(a copy of the approval letter has been sent to the Assistant 527 Editor).”
Line 539: Any role of the funders in the design of the study; in the collection, analyses or interpretation of data; in the writing of the manuscript, or in the decision to publish the results must be declared in this section. If there is no role, please state “The funders had no role in the design of the study; in the collection, analyses, or interpretation of data; in the writing of the manuscript, or in the decision to publish the results”.
References
Formatting of references do not comply with journal requirements. Please check Template and format accordingly. DOI numbers missing in many cases. Journal names should be abbreviated. Mistypes present.
I suppose [7] should not be used, as taxonomy of Felidae was changed a lot, see [8].
For [17], correct link is (I suppose, and it is working) https://www.gov.za/sites/default/files/gcis_document/201512/39468gen1190.pdf
[92] details missing
Reviewer 2 Report
This MS represents the firts attempt to determine the ecology of white lions in terms of spatial behaviour in fenced reserves of South Africa.
The MS is almost well written, but there are some points that need to be addressed before further considerations:
1. First of all, I definitely miss the conservation importance of the white lion, which, as far as I know, is a human-selected variety. Why is it important to study ecology of this taxon? Please be clearer.
2. Figures with maps are unreadable.
3. I know we are talking about lions and not about mice, but your sample size is still low and you need to discuss it. How did you manage pseudoreplications (i.e. same pride)? Please clarify better. Also, how did you consider different size of fenced areas?
4. If I understand well, prey availability is somehow altered by humans. Can you quantify how much this may have affected lion behaviour?
Round 2
Reviewer 1 Report
Manuscript ID: animals-1828784
Home range and movement patterns of reintroduced white lions (Panthera leo melanochaita) in the Kruger to Canyons Biosphere Reserve, South Africa
Authors: Jason A. Turner *, Emma Dunston-Clarke, Inger Fabris-Rotelli, Hans De Iongh
Review, round 2
Authors did requested revision, answering most of my comments. However, they did revision in hurry (answer 20: Due to time constraints, DOI numbers have not been included at this time.). This is no excuse, I think. For sure, there is a mess with Table and Figure numbering, and citing these in the text. Please have some respect to people, reading your work – you take their time.
So there are comments for the further revision. MDPI is flexible with giving authors more time, not the 3, 7 or 10 days which are asked in the standard procedure.
Lines 173–179: this is ONE sentence, Please split it.
Figure 1: please compose both parts sideways.
Lines 202–203: use en dash for the ranges
Table 1: formatted not according Template, there is special style for the Table body. Two columns, GPS coordinates and Reserve size, are the same in both lines, therefore should be given in the text above, e.g., Line 193.
Line 247: (2010/2011 and 2018/ 2020), mistype, remove space after slash
Line 311: correct to (Tables 2–5). I am not sure, if there is no mistake, as Table 4 seems to be the last one.
Tables 2 to 5: formatted not according Template
Figures: wrong numeration, presented not according numbers (e.g., Figure 6 is prior to Figure 4), Figure 5 has no citation in the text
References were not formatted according the Template. Mistypes remain. My previous comment was not answered.
Reviewer 2 Report
Dear authors,
thanks for considering my previous recommendations. Now, the MS can be accepted for publication on Animals.
Author Response
On behalf of myself and my co-authors, thank you to Reviewer 2 for reviewing the manuscript.
Best regards,
Jason Turner